# Hydrodynamic Model Optimization for Marine Tourism Development Suitability in Vicinity of Poso Regency Coastal Area, Central Sulawesi, Indonesia

**Surya Hermawan \*, Edwin Mihardja \*, Devian Aryo Pambudi and Jason Jason**

Civil Engineering Department, Faculty of Civil Engineering and Planning, Petra Christian University, Surabaya 60236, Indonesia
\* Correspondence: shermawan@petra.ac.id (S.H.); edwin.mihardja@gmail.com (E.M.)

**Abstract:** Poso regency, Central Sulawesi, Indonesia, has a coastal area that has the potential to be developed for marine tourism. It is expected that marine tourism can bring socio-economic impact to the community. This research was conducted with the objective of assessing the suitability of the area to be developed as a marine and coastal tourism site to provide benefits to the coastal community. A hydrodynamic model was used in this research for coastal area mapping. As an approach, Analysis Hierarchy Process (AHP) was utilized, whose parameters consist of depth, coast type, coast width, brightness, current speed, water base materials, observation of dangerous biota, and availability of fresh water. Based on the overall mapping area of 98,644 ha, the research results show that the area that can be utilized is 7979 ha with a 'very suitable' category, while there is a further area of 1045 ha which can still be classified in the 'suitable' category.

**Keywords:** hydrodynamic model; marine and coastal tourism; Analysis Hierarchy Process (AHP)

## 1. Introduction

Coastal areas are considered as essential from an environmental and economic standpoint [1]. These two standpoints have detrimental consequences to the coastal communities. Therefore, it is important to manage those environments' resources sustainably [2]. Typically, resources of a coastline destination play a significant role in luring investments and infrastructure related to tourism. This is because coastal areas typically have a variety of natural resources in addition to cultural treasures [3]. According to Bob et al. (2018), coastal and marine tourism (CMT), which is a component of the ocean economy, has enormous development potential that can support sustainability and the creation of jobs [4].

Coastal tourism is an activity that involves visitors, locals, and the places they want to go, especially the coastal environment and its natural and cultural elements. [5]. Marine tourism and coastal tourism are two of the oldest forms of travel, and these environments are highly sought-after destinations for tourists [6]. Therefore, marine and coastal tourism is one of the largest and fastest-growing divisions of the tourism business. According to projections, 8.6 million people would be employed by marine and coastal tourism by 2030. Hence, it makes up 26 percent of the entire global ocean economy [7]. Sustainable development is impacted by coastal and marine tourism in all its aspects. The natural ecosystems of coastal resorts are altered by poorly managed tourism activities in terms of their environmental impact. The demand for natural resources has grown dramatically because of widespread tourism. The coasts are overrun by mass tourism, which damages local ecosystems and has a negative impact on local species in coastal ecosystems. Additionally, induced land-use change contributes to coastal artificialization and transport-related air and noise pollution. Through the creation of negative externalities that are frequently uncompensated, this condition affects the welfare of residents and local communities [8].



Indonesia, as an archipelagic country with 17,499 islands and a total coast length of almost 81,000 km, as well as being the largest nation in the Indian Archipelago, is offering tremendous potential for coastal and marine tourism. The tourism industry has helped to enhance the country's overall Gross Domestic Product (GDP) since Indonesia is one of the most desirable tourism destinations for tourists around the world [9]. Development of the tourism industry in Indonesia has been planned to lessen poverty and protect the environment and natural resources [10]. The main challenge for the coastal planner in designing the development of coastal tourism in remote areas is a limited number of measurable data, along with the capacity building of the people.

Poso Regency's 4013 km of coastline offers a lot of potential for marine ecotourism development (see Figure 1). However, the local coastal communities are not utilizing it optimally. If the natural resource potential is explored to its full potential, it can help to bring positive socio-economic impact to local communities. Most of the coastal communities in Indonesia's coastal regions live in poverty, with a per capita income below World Bank norms. Hence, the tourist industry is one of the key areas where it is expected to make a significant contribution to the efforts to improve the existing economy in coastal communities. It can be stated as well that coastal regions have a high economic value that includes both tangible and intangible economic benefits. Based on the tourist visitation report issued by Central Sulawesi Tourism Office in 2020, there were increases in foreign tourist numbers in 2021 and 2022, with Poso regency being one of the main attractions for foreign tourists besides Tojo Una-Una and Palu Regency. As tangible benefits, there is a potential increase in regional Gross Domestic Product (GDP) in Central Sulawesi province from USD 270,000 in 2022 to USD 521,000 in 2023. If the coastal area in Poso regency is further developed as an area of coastal tourism, it will increase Poso's GDP contribution to the tourism sector in Central Sulawesi Province. The tourism sector itself has contributed on average between 3 and 3.5% to the total provincial GDP. Intangible benefits will be found in the infrastructure development for tourism and maintenance of the ecosystem in the tourism area. Hence, it can support the local economic growth in coastal areas [11].

Poso Regency (see Figure 1) was chosen as a field for this research because it has a lot of natural resources. On the contrary side, these natural resources are not currently being utilized to their full potential by the coastal communities. Previous studies for developing tourism in Poso Regency were mostly focused on land-based tourism. Balingki (2015) has proposed a strategy for development of coastal tourism in Lake Poso by analyzing internal and external factors, i.e., tourism potential, accessibility as internal factors, and government support as external factors [12]. A further study on the implementation of strategy development for coastal tourism in Lake Poso was conducted by Rembang et al. (2017) to evaluate the effectiveness of the local government tourism office for supporting the development program [13]. Like the research above, a development strategy for tourism in Wera Saluopa waterfall was conducted by Abidjulu (2019), finding that in combination with attractions, such as a cacao plantation, it is possible to bring added value [14].

As the first step in the feasibility study for developing the marine or coastal tourism area, an engineering approach is used in this work. This research will be conducted by building a hydrodynamic model to gather information such as water level conditions, current characteristics, wind speed, and tides to determine the oceanographic conditions in the Poso Regency area. With the assistance of these data, it should be possible to decide what actions should be taken in the future to develop the Poso coastal area in terms of tourism development, marine culture, and other areas. It is expected that this research will make a significant contribution to the community in and its surroundings in Poso Regency, particularly to the community in the coastal area, by providing information, education, and applications for optimizing natural resources. Education to the community is needed to prepare its readiness for foreign and domestic tourism development. Unfortunately, an unstable environment is still susceptible to violence. This poses one of the largest threats to Poso Regency's tourism industry [15].

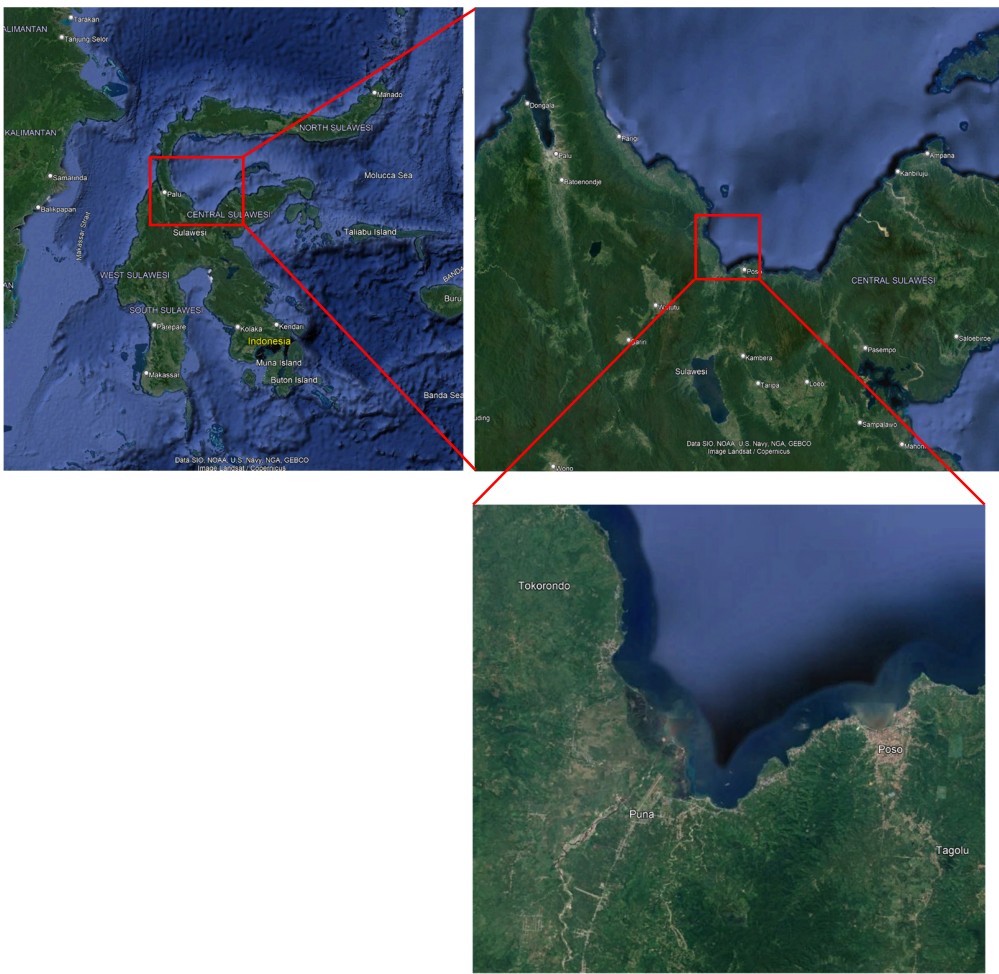

**Figure 1.** Research field area, coastal area in Poso Regency in Central Sulawesi, Indonesia.

## 2. Materials and Methods

To create this hydrodynamic model simulation, Poso Beach, which is in Poso Regency, Central Sulawesi, Indonesia, was utilized as the setting point. The model was run for 2 weeks. The first step was preparing and determining observation points around the coastal area of Poso Regency. Secondly, the data needed for research were prepared from the General Bathymetric Chart of the Ocean (GEBCO) for bathymetry data, Copernicus for wind data, and International Hydrographic Organization (IHO) for water level data. The hydrodynamic model was then assessed for its validity and accuracy with Root Mean Square Error (RMSE) method. It was compared also with the data on land, air, and sea observations from Copernicus. The study used wind data, which were collected from the last ten years, from 1 January 2011 to 31 March 2021. The gathered wind data were then processed to create a wind rose diagram (see Figure 2).

### 2.1. Hydrodynamic Model Using Delft3D

The Delft3D model can simulate currents, waves, sediment movement, water quality, morphological development, and ecology in the modeled area, while simulating the original circumstances in rivers, estuaries, and coastal areas. In practice, this application can be coupled with ArcGIS 10.8 and needs several other supporting programs, such as MATLAB. The hydrodynamic modeling program Deflt3D uses numerical techniques to process input data. Astronomical equations (see equation) are used to drive the model by supplying details regarding its boundary conditions (the land border).

The general formula for the astronomical tide which drives the hydrodynamic numerical model is:

$$H(t) = Ao + \sum_{i=1}^{k} Ai\ Fi \cos(\omega it + (Vo + u)i - Gi)$$

H(t) = Water level at time t
Ao = Mean water level over a certain period
k = Number of relevant constituents
i = Index of a constituent
$A_i$ = Local tidal amplitude of a constituent
$F_i$ = Nodal amplitude factor
$\omega_i$ = Angular velocity
(Vo + u) = Astronomical argument
$G_i$ = Improved kappa number (=local phase lag)

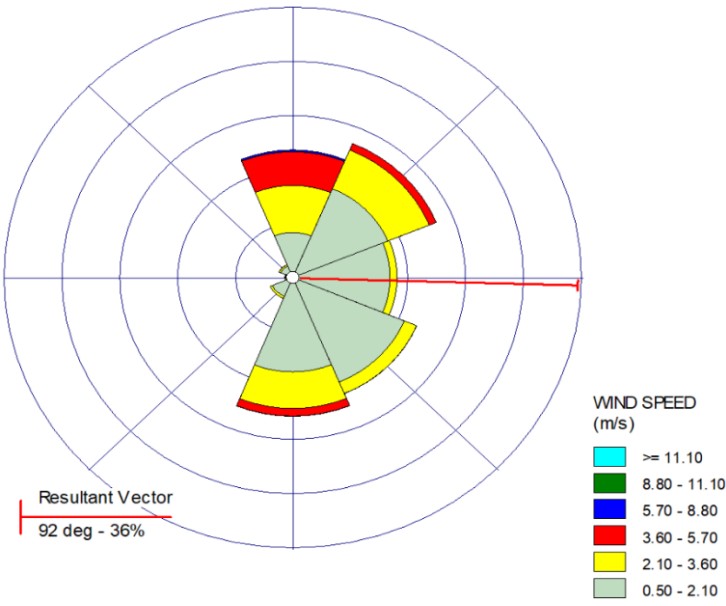

**Figure 2.** Wind Rose Diagram.

The primary data are bathymetry data from GEBCO Bathymetry Data, or sounding data from Deeper Smart Sonar. This is to ensure the accuracy of the simulation results. It can serve as a basic reference to see the present and forecast the future dynamic movement of water.

The Shallow Water Equation (SWE) is calculated using the Delft3D-Flow 4.04.02 software in Delft3D with two different variables, namely the velocity and the height. These variables are then projected onto a line with two directions—horizontal and vertical—or what the Delft3D-Flow application refers to as a grid. The Navier–Stokes equation is used in the open source Delft3D 4.04.02 software system's calculations. The Navier–Stokes formula is described in the following equation [16].

$$\frac{\partial \rho}{\partial t} + \frac{\partial(\rho u)}{\rho x} + \frac{\partial(\rho v)}{\rho y} + \frac{\partial(\rho w)}{\rho z} = 0$$

x, y, z = Coordinates
u, v, w = Speed components
$\rho$ = Density

### 2.2. Root Mean Square Error (RMSE)

When evaluating the performance of models in studies like meteorology, air pollution, and climate research, the Root Mean Square Error (RMSE) approach is frequently employed as a standard statistical metric. Numerous researchers in geosciences employ the RMSE approach as a common metric to evaluate the model's level of error [17]. Consequently, using the RMSE approach, the hydrodynamic model's error can be calculated using the RMSE formula:

$$RMSE = \sqrt{\frac{\sum_{i=1}^{N}(Predicted\ i - Actual\ i)^2}{N}}$$

Predicted$_i$ = Predicted value data
Actual$_i$ = Actual simulation
$N$ = Number of data

RMSE has been used as a benchmark for assessing the hydrodynamic model's performance [18,19]. If the RMSE value is within or less than 0.1, the modeling has been done correctly and could be considered as accurate. On the other hand, if the RMSE value is more than 0.1, the required data collecting phase to support the simulation must be repeated.

The validated hydrodynamic model can be used as a supplement to the online collected data as well as being utilized for mapping the suitability of a coastal area. The suitability for tourism development is mapped with ArcGIS technology. Only were the waterways surrounding the Poso Regency area used for the mapping.

### 2.3. Analysis Hierarchy Process (AHP)

The Analysis Hierarchy Process (AHP) approach can be used to process a suitability mapping from each of the above suitability parameters. According to Figure 3 [20], this strategy involves estimating values using an analytical framework. As illustrated in Figure 3, the suitability parameter data were divided into three categories: highly suitable, suitable, and unsuitable. Following reclassification, the data were compiled and once more evaluated against the preexisting parameters as shown in Figure 3, resulting in the creation of the suitability map's results [21].

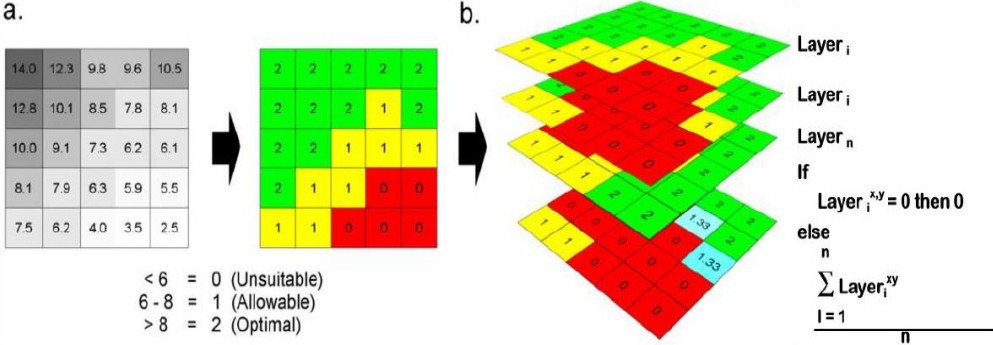

**Figure 3.** Value summation scheme: (**a**) Reclassification of Parameter Data (**b**) Estimation of Some Parameter Data for Final Results.

## 3. Results

### 3.1. Model Simulation Using Delft3D

A square grid and bathymetry data from *GEBCO* were used for the hydrodynamic model simulation. The results from several generated simulations with different grid sizes and Manning's roughness is shown in Table 1.

**Table 1.** Delft3D model simulation.

| No. | Model | Grid (Meter) | Bathymetry | Manning's Roughness | Time Step (Minutes) | Simulation Duration (Days) | Desc. |
|-----|-------|--------------|------------|---------------------|---------------------|----------------------------|-------|
| 1 | A | 200 | GEBCO | 0.05 | 60 | 365 | Failed |
| 2 | B | 1200 | GEBCO | 0.05 | 60 | 365 | Failed |
| 3 | C | 1600 | GEBCO | 0.05 | 5 30 60 | 30 | Failed |
| 4 | D | 5550 | GEBCO | 0.05 | 5 30 60 | 14 | Successful |
| 5 | E | 5550 | GEBCO | 0.05 0.033 0.025 | 5 | 14 | Successful |

All model simulation experiments were successful with different Manning's roughness numbers in model E. Therefore, Model E is the model used for further research.

### 3.2. Validation and Verification of Simulated Hydrodynamic Model

The RMSE approach was used to test and verify the model between the simulation results and the results of the measuring station (Delft Dashboard). The water level elevation data from the model in Figures 4 and 5 were used in the RMSE formula. The IHO monitoring station relates to the Poso station, which can be found in the Delft Dashboard program (see Table 2).

Furthermore, this model was used as supporting data for mapping the suitability of coastal area development in Poso Regency with the ArcGIS application.

### 3.3. Coastal Area Development Mapping with ArcGIS

The suitability mapping of coastal marine tourism in this study was based on the evaluation of depth, brightness, slope, current speed, and wave height as criteria in Table 3.

Depth mapping is shown in Figure 6. More than 90% of the area, about 88,780 ha, was marked as unsuitable, especially around the center of the map. These areas are unsuitable because the depth is around 6 to 10 m, which is considered dangerous for tourism activity. However, it can be seen that areas around the coast are marked as very suitable since these areas have 0 to 3 m depth. This is considered acceptable and safe for tourism activity.

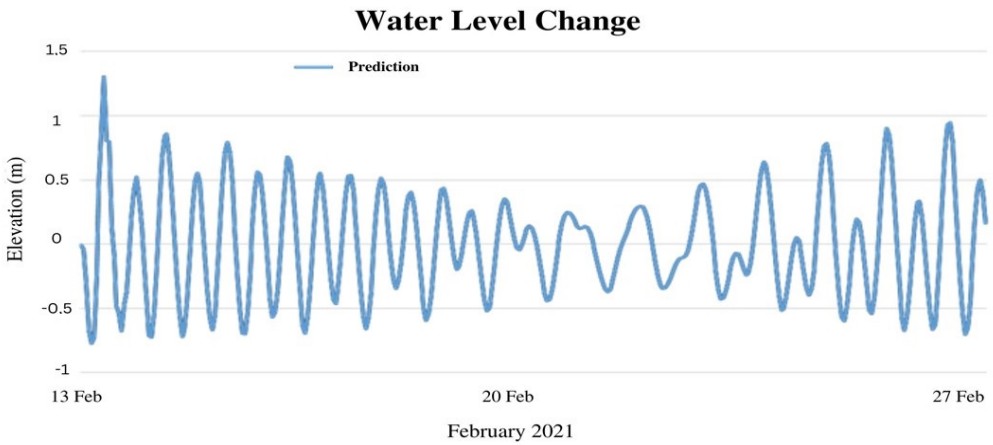

**Figure 4.** Hydrodynamics Model Results in the Form of Water Level Change Graph.

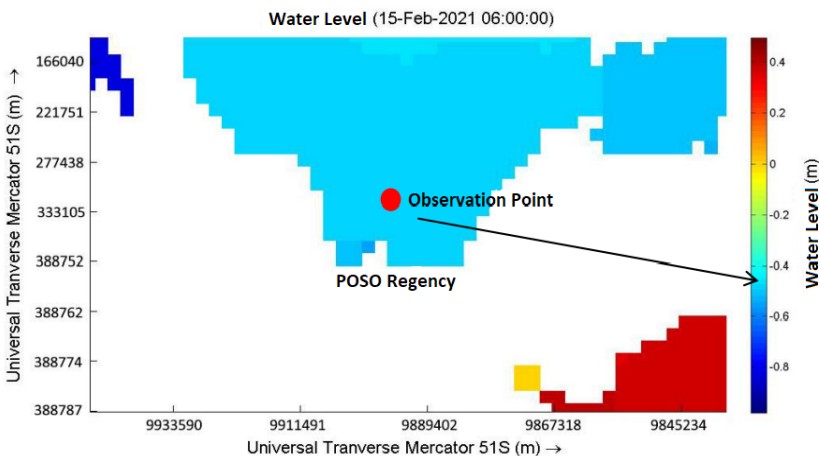

**Figure 5.** Dimensional Simulation Results of Water Level Change (Elevation).

**Table 2.** Root Mean Square Error Calculation Data Model E Manning's Roughness 0.05.

| POSO | | Jumlah Data (n) | 360 |
|---|---|---|---|
| **Dote and Time** | **Predicted (m)** | **Actual (m)** | **(Predicted − Actual)^2** |
| 2021-02-13 01:00:00, | −0.00660431 | −0.266 | 0.067286124 |
| 2021-02-13 02:00:00, | −0.0402799 | −0.42 | 0.144187354 |
| 2021-02-13 03:00:00, | −0.308369 | −0.489 | 0.032627558 |
| 2021-02-13 04:00:00, | −0.667691 | −0.446 | 0.049146899 |
| 2021-02-13 05:00:00, | −0.772871 | −0.29 | 0.233164403 |
| 2021-02-13 06:00:00, | −0.727247 | −0.051 | 0.457310005 |
| 2021-02-13 07:00:00, | −0.240389 | 0.221 | 0.212879809 |
| 2021-02-13 08:00:00, | 0.439049 | 0.464 | 0.000622552 |
| 2021-02-13 09:00:00, | 0..86659 | 0.623 | 0.059336088 |
| 2021-02-13 10:00:00, | 1.30275 | 0.661 | 0.411843063 |
| 2021-02-13 11:00:00, | 0.80441 | 0.57 | 0.054948048 |
| 2021-02-13 12:00:00, | 0.796764 | 0.374 | 0.1787294 |
| 2071-07-13 13:00:00, | 0.123352 | 0.12 | $1.12359 \times 10^{-5}$ |
| 2021-02-13 14:00:00, | −0.11178 | −0.132 | 0.000408848 |
| 2021-02-27 19:00:00, | −0.127693 | 0.054 | 0.033012346 |
| 2021-02-27 20:00:00, | 0.206054 | 0.22 | 0.000194491 |
| 2021-02-27 21:00:00, | 0.430109 | 0.306 | 0.015403044 |
| 2021-02-27 22:00:00, | 0.499308 | 0.285 | 0.045927919 |
| 2021-02-27 23:00:00, | 0.401124 | 0.159 | 0.058624031 |
| 2021-02-28 00:00:00, | 0.166415 | −0.038 | 0.041785492 |
| TOTAL (∑) | | | 12.249 |
| The Root Mean Square Error (RMSE) | | | 0.184 |

**Table 3.** Marine Tourism Suitability Score Beach Category [22,23]. Reprinted/adapted with permission from Yulianda, F, Seminar Sains; published by Departemen Manajemen Sumberdaya Perairan, Institut Pertanian Bogor, 2010.

| Parameter | Unit | Very Suitable | Suitable | Unsuitable |
|---|---|---|---|---|
| Water depth | m | 0–3 | >3–6 | >6–10 |
| Type of coast | | White sand | White sand with coral | Black sand, coral, and stiff |
| Wide of coast | m | >15 | <10–15 | 3–9 |
| Bed coast material | | Sand | Sandy coral | Sandy mud |
| Current speed | m/s | 0–0.17 | 0.17–0.34 | 0.34–0.51 |
| Coastal slope | o | <10 | 10–25 | >25–45 |
| Water clarity | % | >10 | >5–10 | 3–5 |
| Coastal closing area | | Coconut tree, open area | Scrubs, lowland, savanna | High scrubs |
| Dangerous biota | | No | Jellyfish, sea urchin | Nani's fur, stingray |
| Freshwater | Km | <0.5 | >0.5–1 | >1–2 |

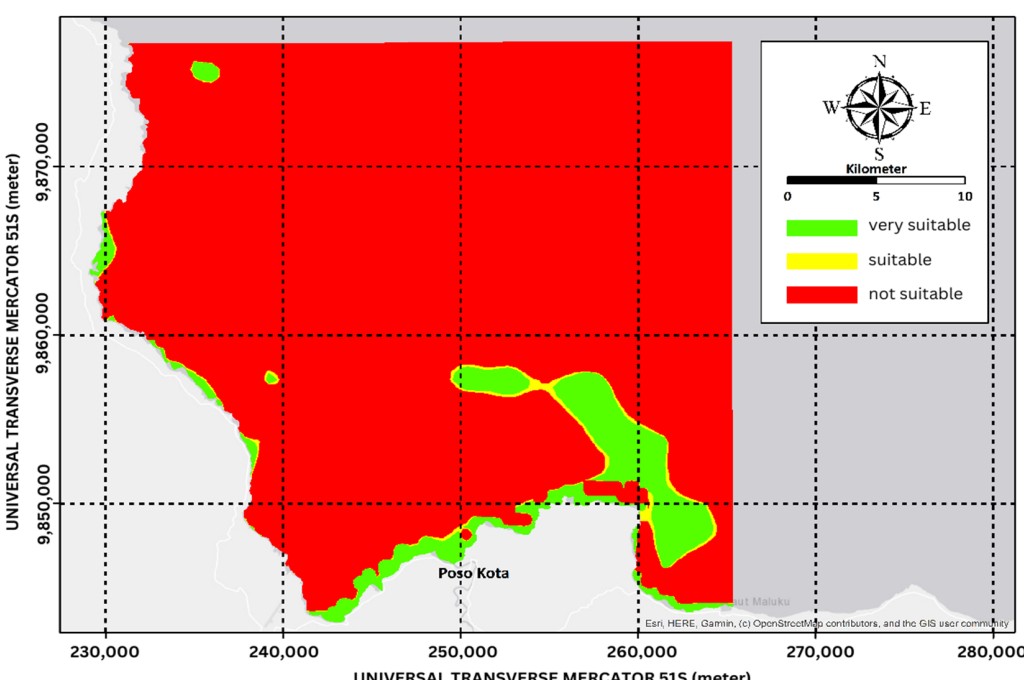

**Figure 6.** Beach Category for Marine Tourism Suitability Map Based on Depth.

Based on the current speed category, all areas on the suitability marking are marked green as shown in Figure 7 as very suitable for marine tourism. It can also be observed that the current speed in this area can be considered low, at about 0.17 m/s, which makes it suitable for tourism.

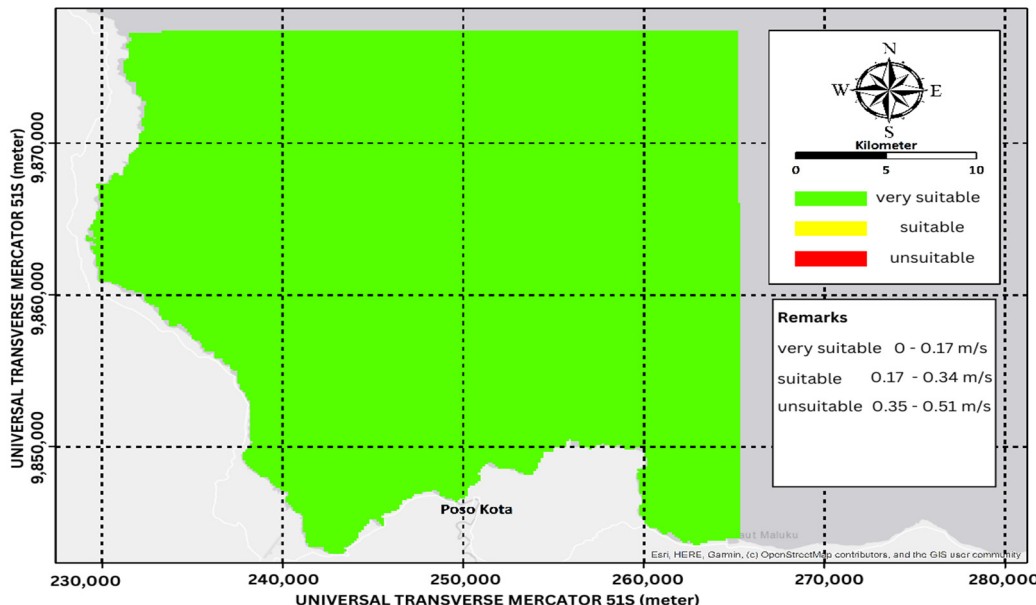

**Figure 7.** Coastal Area Category Marine Tourism Suitability Map Based on Current Speed.

As shown in Figure 8, about 70% of the mapped area, about 69,050 ha, was marked as unsuitable with red color. It indicates that those areas which have less than 5% water clarity, about 4952 ha, are both unpleasant and dangerous for marine tourism. There are only certain areas that are categorized as very suitable and suitable. These areas are shown in green color as very suitable category and yellow color as suitable category.

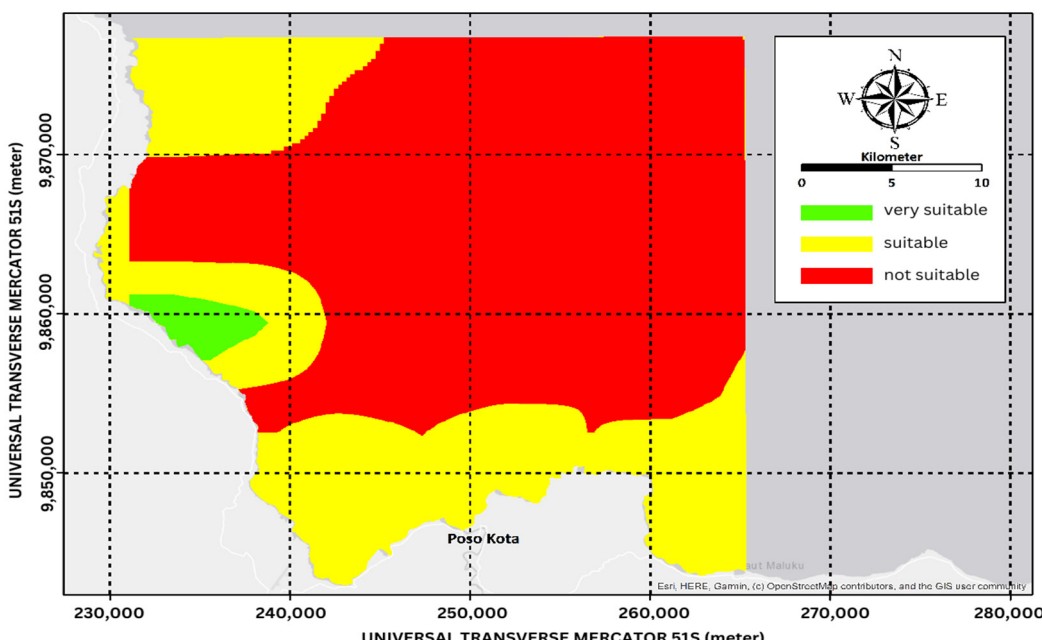

**Figure 8.** Beach Category Marine Tourism Suitability Map Based on Brightness.

Based on slope category, as displayed in Figure 9, most of the mapped areas are marked as very suitable with less than 10° slopes. There are some small areas, at 2% of the total area (1972 ha), with yellow color are considered as suitable. Like the slope category, it can be seen in Figure 10, all mapped areas are considered very suitable for marine tourism based on wave height.

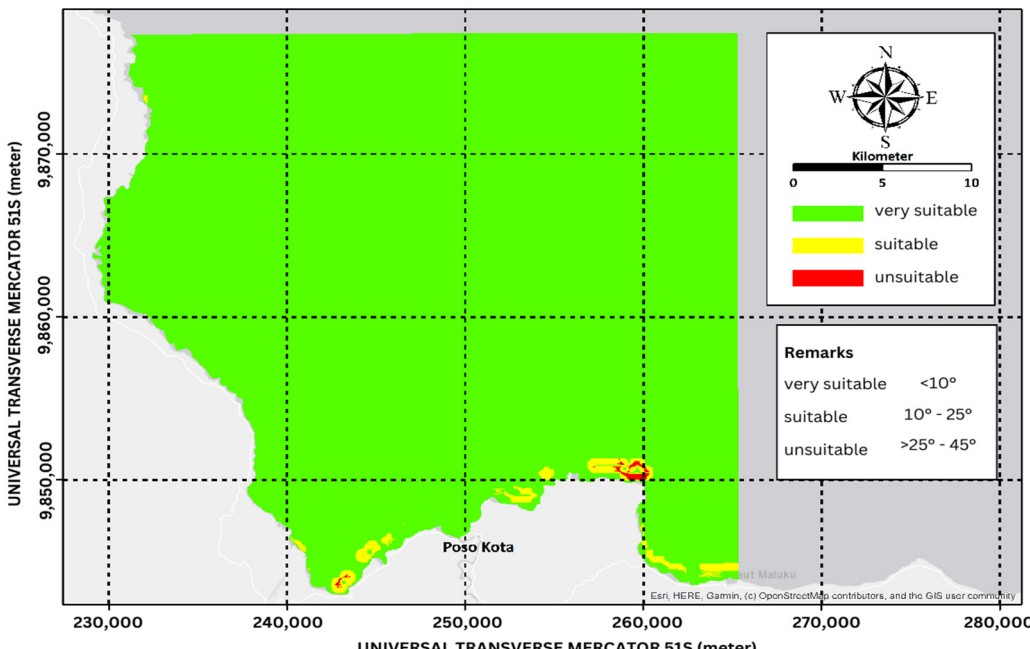

**Figure 9.** Beach Category Marine Tourism Suitability Map Based on Slope.

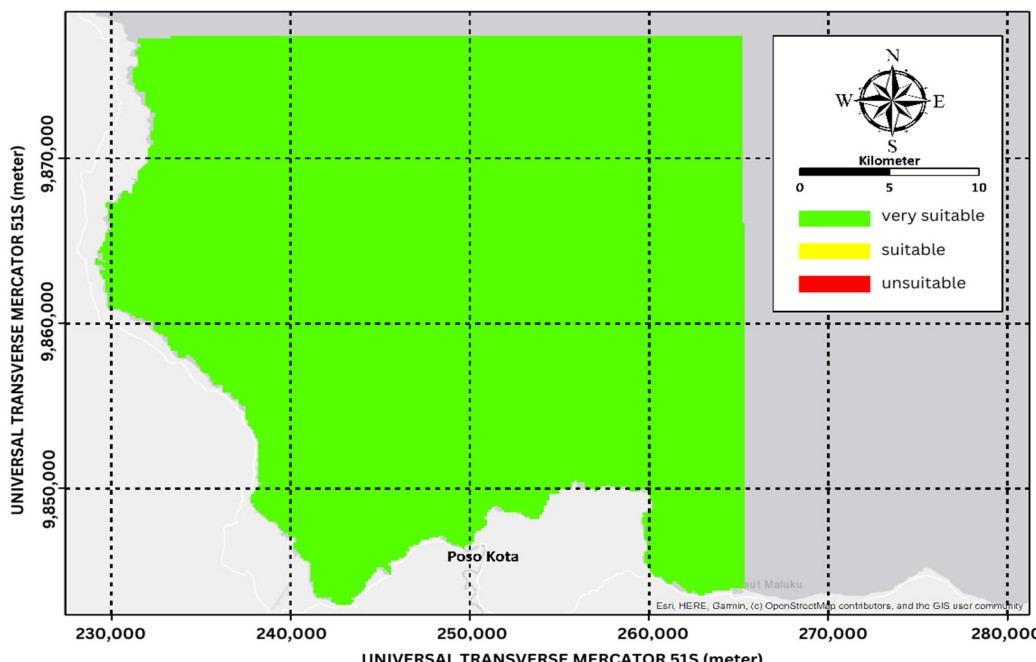

**Figure 10.** Beach Category Marine Tourism Suitability Map Based on Wave Height.

The overall results, by using the AHP approach and considering the depth, current speed, brightness, slope, and wave height factors, indicate that most of the coastal areas in Poso Regency are suitable for marine tourism. It can be seen from Figure 11 that those areas (7979 ha) marked with green color are very suitable for marine tourism, while there are also 1045 ha marked in a yellow color that can be considered as suitable as well. Some coastal areas, and most of the open water areas, in red color are marked as unsuitable for marine tourism because they have depth more than 10 m and low water clarity.

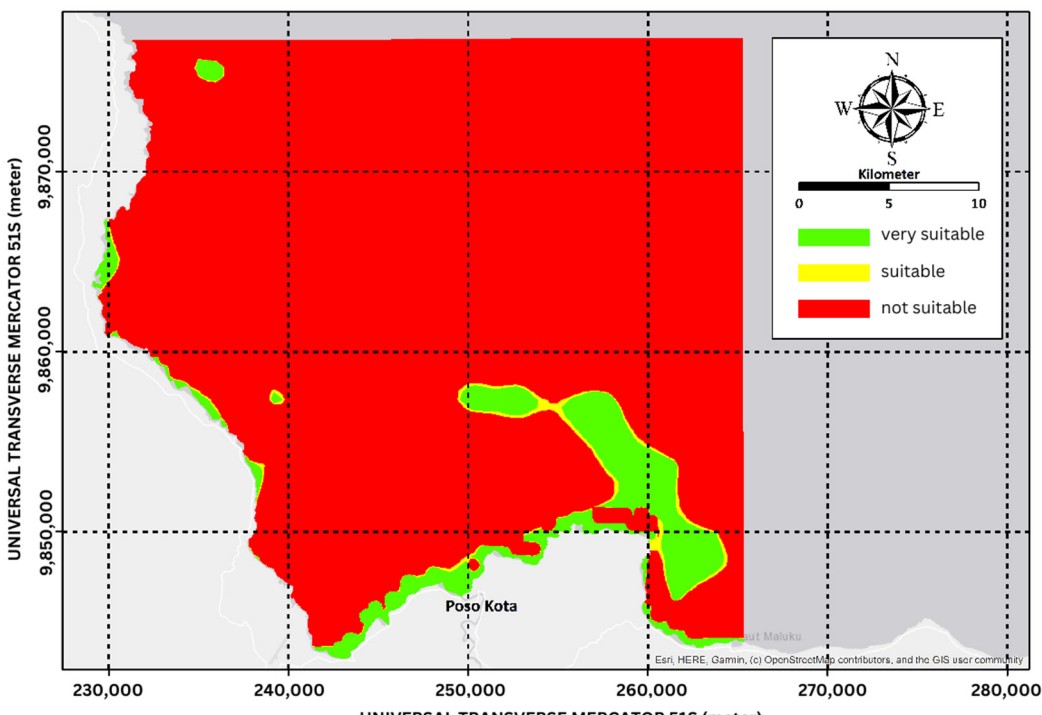

**Figure 11.** Result of Beach Maritime Tourism Suitability Map.

## 4. Discussion

Data simulations were run from 13 to 28 February 2021 for a period of 2 weeks. Model A simulations were run with a 200 m grid, Manning's roughness of 0.05, and a time step of 60 min. The simulation failed because the grid was unstable. In Model B, the grid was improved by increasing the grid size to 1200 m and orthogonalizing the grid. However, Model B still experienced errors. The grid was then improved again in Model C by increasing the grid size and using a time step of 5, 30, and 60 min. It turned out that Model C still has errors. The Manning's roughness could not be increased because the largest Manning's roughness limit in the model is 0.05. Model D was created with 5550 m grid and 0.05 as the Manning's roughness value. It turned out to be successful with 3 different time steps.

Model E was created with the same grid as Model D, but with 3 different Manning's roughness values. With this modification of Model D, all parameters in Model E were successfully simulated. Model E, with a Manning's roughness number of 0.05, has an RMSE value close to 0.1, namely 0.184. It can be concluded that Model E is accurate enough to describe the original condition of the waters in Poso Regency. The simulated hydrodynamic data from Model E can be applied in Analysis Hierarchy Process (AHP) to determine the suitability of the areas which can be developed for marine tourism.

The AHP mapping result shows that Poso Regency's coastal areas have suitable depth for marine tourism activity with less than 3 m of depth. Compared to the Pantar Strait area, which has a depth of 1.1–2.6 m [24], Poso Regency's coastal areas are more suitable for swimming and bathing. Based on current speed, with a maximum current speed of 0.17 m/s, it is marked as very suitable [25]. In the water clarity category, the AHP mapping showed many varieties of marks along the areas. Even though it shows lower water clarity compared to other coastal areas, it was still marked as very suitable. The coastal area in Poso Regency is considered as flat beach, which has a slope of less than 10°. This will be an advantage in the safety and convenience factors for marine tourism.

## 5. Conclusions

The results of the beach category marine tourism suitability map, based on the mapped region with a total area of 98,466 ha, have revealed that 7979 ha is 'very suitable', while 1045 ha is 'suitable' for marine tourism development. However, it is noted that mapping of dangerous animals cannot be acquired using internet stations. To meet all these requirements, either regional data provided by the local government or live data is needed. Only are four out of ten suitability factors—depth, brightness, slope, and current speed—included in mapping the suitability of coastal marine tourism in this research.

From the engineering side, it is expected that this research will contribute to the development of coastal tourism by introducing an engineering method to the local government tourism office on how to conduct a feasibility study in determining a suitable coastal tourism area. The next step will be focused on the preparation of a tourism development plan and its supporting infrastructure based on this research. It is needed to also prepare the SOP for QHSE matter in the development plan to avoid any dangerous situation such as drowning hazard and determining "no go" areas. As the COVID 19 pandemic is still lurking, there must be an SOP which regulates the health regulation there, distance keeping, obligation to wear a mask, etc.

**Author Contributions:** Conceptualization, S.H.; Methodology, D.A.P. and J.J.; Software, D.A.P. and J.J.; Validation, D.A.P. and J.J.; Formal analysis, D.A.P. and J.J.; Resources, S.H.; Writing—original draft, E.M.; Writing—review & editing, E.M.; Supervision, S.H. and E.M. All authors have read and agreed to the published version of the manuscript.

**Funding:** This research was funded by Faculty of Civil Engineering and Planning Petra Christian University Surabaya Indonesia and Institute of Research and Community Outreach Petra Christian University.

**Conflicts of Interest:** The authors declare no conflict of interest.

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
