# Peer review of "Hydrodynamic Model Optimization for Marine Tourism Development Suitability in Vicinity of Poso Regency Coastal Area, Central Sulawesi, Indonesia"

_sustainability, doi:10.3390/su15043150_

Round 1
Reviewer 1 Report
Dear Author(s),
The topic of the manuscript is of interest and relevance, and it offers several references to support it. The research on increasing anthropogenic pressure in certain natural environments –especially on vulnerable ecosystems– and coastal management initiatives are key issues in the sustainability context. The topic discussed in your document could provide an important contribution to this research area. It also provides an interesting case study. I will therefore recommend its publication in Sustainability.
Author Response
Thank you for the review and we glad that we can contribute a practical approach from engineering side so that the method can be used further in the feasibility study / assessment before developing a marine / coastal tourism area. It is expected also that there will be an improvement in this method in the future and it can help local government or investors to develop a tourism plan.
In the attachment you may find also the response in pdf file as per guidance

Reviewer 2 Report
This research was conducted with the objective of assessing the suitability of the area to be developed as a marine and coastal tourism site to provide benefits to the coastal community. The article generally concerns the physical parameters of the examined area, whether a tourist can swim there or not. Such research is also appropriate to ensure the safety of tourists. I would rather be interested in how this marine tourism will affect the local community. What to do so that this community does not become one of many that have lost their local, traditional, and cultural character, and become one of those global tourist villages that meet the needs of tourists, but forget about the inhabitants. How to protect such places from a tourist disaster?
I have missed a description of previous research in this area. Has this research been done before by other authors and where? How has it influenced the development of tourism in the area studied?
What are the conclusions of this study for the local community, what next?
Author Response
Question:
This research was conducted with the objective of assessing the suitability of the area to be developed as a marine and coastal tourism site to provide benefits to the coastal community. The article generally concerns the physical parameters of the examined area, whether a tourist can swim there or not. Such research is also appropriate to ensure the safety of tourists. I would rather be interested in how this marine tourism will affect the local community. What to do so that this community does not become one of many that have lost their local, traditional, and cultural character, and become one of those global tourist villages that meet the needs of tourists but forget about the inhabitants. How to protect such places from a tourist disaster?
Answer:
Poso regency is a regency in Central Sulawesi with about 245000 inhabitants and total area of 7112,25 km2. Most of the population reside in coastal areas. As current situation, they are developing seaweed cultivation in the coastal area. The famous tourist areas are Poso Lake, Lore Lindu National Park and Tambing Lake which is famous for endemic bird paradise and visited by foreign tourists. Poso regency has in general a great potential to be developed and this has been started also by local government. Marine tourism can be developed besides land tourism and seaweed cultivation. It will impact the economic growth for the community and bring an added value if marine tourism is developed, i.e., creating employment to the residents by increasing tourism industry and bringing local and national income from foreign tourism.
Answer:
To keep their local traditional and cultural character, residents (supported by the local government) could introduce the local tradition, culture, and discipline (if necessary) to the foreign tourist so that they can enjoy the diversity of the culture and adapt with the local culture, e.g., introducing their local food / dishes, and local wisdom / local culture and product. It must be maintained and supported by local government so that the community doesn’t lose its character and culture because of globalization. We feel also the same that a lot of residents in tourism places lost their cultural identity which we can consider as tourist disaster. To protect this from tourist disaster, a disciplinary action according to local wisdom must be established so that foreign tourists can’t do anything that they want e.g., establishment of local police in village administration, such as Pecalang which we can find in Bali.
Question:
I have missed a description of previous research in this area. Has this research been done before by other authors and where? How has it influenced the development of tourism in the area studied?
Answer:
According to the best of our knowledge, there is limited local research article specifically developing marine / coastal tourism in Poso regency. Most of the local research (in Bahasa) which we found is the development of tourism industry in Poso (land-based tourism, such as Poso Lake, waterfalls, etc.) and its rebranding for tourism. It is expected from the research that there will be investor which can cooperate with the local government tourism office to develop the tourism potential in Poso regency.
Question:
What are the conclusions of this study for the local community, what next?
Answer:
From engineering side, research team expects that the article will contribute to the development of marine / coastal tourism by introducing a simple method to local government tourism office how to conduct a feasibility study / assessment in determining a suitable coastal tourism area. The next step will be focusing on the preparation of a tourism development plan based on this research result. Of course, it is needed to also prepare the SOP for QHSE matter in the development plan to avoid any dangerous situation such as drowning hazard and determining “no to go” area. As COVID 19 pandemic is still lurking, there must be a SOP which regulates the health regulation there, distance keeping, masker obligation, etc.
